# Toward Morphologic Atlasing of the Human Whole Brain at the Nanoscale

Wieslaw L. Nowinski

Sano Centre for Computational Personalised Medicine, 30-054 Krakow, Poland; w.nowinski@sanoscience.org

**Abstract:** Although no dataset at the nanoscale for the entire human brain has yet been acquired and neither a nanoscale human whole brain atlas has been constructed, tremendous progress in neuroimaging and high-performance computing makes them feasible in the non-distant future. To construct the human whole brain nanoscale atlas, there are several challenges, and here, we address two, i.e., the morphology modeling of the brain at the nanoscale and designing of a nanoscale brain atlas. A new nanoscale neuronal format is introduced to describe data necessary and sufficient to model the entire human brain at the nanoscale, enabling calculations of the synaptome and connectome. The design of the nanoscale brain atlas covers design principles, content, architecture, navigation, functionality, and user interface. Three novel design principles are introduced supporting navigation, exploration, and calculations, namely, a gross neuroanatomy-guided navigation of micro/nanoscale neuroanatomy; a movable and zoomable sampling volume of interest for navigation and exploration; and a nanoscale data processing in a parallel-pipeline mode exploiting parallelism resulting from the decomposition of gross neuroanatomy parcellated into structures and regions as well as nano neuroanatomy decomposed into neurons and synapses, enabling the distributed construction and continual enhancement of the nanoscale atlas. Numerous applications of this atlas can be contemplated ranging from proofreading and continual multi-site extension to exploration, morphometric and network-related analyses, and knowledge discovery. To my best knowledge, this is the first proposed neuronal morphology nanoscale model and the first attempt to design a human whole brain atlas at the nanoscale.

**Keywords:** human brain; morphology; modeling; atlasing; neuron; nanoscale; big data; high-performance computing

## 1. Introduction

In the last few years, we have witnessed an immense explosion of human brain-related efforts that are big, advanced, and well-funded projects, initiatives, and/or long-term national and multi-national research programs, such as *The Allen Brain Atlas* to map gene expression [1], *The Big Brain* to obtain ultra-high resolution neuroimages [2], *The Human Connectome Project* to map structural and functional connections [3], *The BRAIN Initiative* (*Brain Research through Advancing Innovate Neurotechnologies*) to develop technology advances in neuroscience discovery [4], *The Blue Brain Project* to simulate neocortical micro-circuitry [5], *The Human Brain Project* to create a research infrastructure to study the human brain [6], and *SYNAPSE* (*Synchrotron for Neuroscience—an Asia-Pacific Strategic Enterprise*) to map the entire human brain at the sub-cellular level using synchrotron tomography [7], among others. These large-scale endeavors immensely increase our knowledge about the human brain and boost the development of novel and more powerful brain atlases. This tremendous brain atlas development proceeds at multiple scales and in numerous directions [8].

The brain atlas has been understood and defined in various ways depending on its nature and application; for instance, as a collection of micrographs or schematic drawings

of brain sections with identified anatomic structures from one or a few brains [9], a large-scale neuroimaging database that captures the mean and variance in the population [10], a tool for localizing experimental data and planning experiments [11], and a teaching file of brain anatomy and function comprising a three-dimensional (3D) brain image with a set of labels [12].

To comprehensively render the vast potential of the human brain atlas, I have defined it as "a vehicle to gather, present, use, share, and discover knowledge about the human brain with a highly organized content, tools enabling a wide range of its applications, massive and heterogeneous knowledge database, and means for content and knowledge updating and growing by its user" [13]. In this study, the concept of the considered nanoscale brain atlas is aligned with this definition.

A human whole brain atlas at the nanoscale has not yet been created. When considering the construction of such an atlas, there are at least four key challenges including (1) the feasibility of acquiring nanoscale data for the human whole brain in a reasonable time and cost, (2) the feasibility of storing and processing these data, (3) the modeling of the human brain at the nanoscale and here we focus on morphology, and (4) the design of a nanoscale brain atlas.

Neuroimages can be acquired at different spatial scales and of various modalities, including macroscopic from magnetic resonance imaging and computed tomography, mesoscopic from optical imaging, and micro- and nanoscopic from electron microscopy. So far, the entire human brain anatomy has been modeled from neuroimaging at the macroscale by several groups [14–17]. However, acquiring the human whole brain data at the nanoscale requires, depending on an employed modality, a prohibitively long time. For instance, to image the human whole brain by applying the same brain imaging procedure as was used for Drosophila's brain [18] would require an estimated 17 million years [19]. Fortunately, thanks to progress in imaging, the acquisition time keeps decreasing. A promising imaging modality is synchrotron X-ray tomography [7,20–22] capable of decreasing the human whole brain acquisition time at the sub-cellular level to a few years [7]. In particular, by employing phase-contrast synchrotron tomography, human whole brain imaging was completed in about 16 h at a 25 μm spatial resolution [21]. This level of spatial resolution is sufficient to image neuronal cell bodies, but it is not satisfactory enough to demonstrate the complete neurons with their dendrites and axons as well as the synapses that functionally connect the neurons.

It is well known that neuroscience requires big data and high-performance computing [23] even for small animals [24]. I have earlier evaluated the bigness of some of these neurodata and just to store the morphology of the extracted neurons for the entire human brain, a storage on a petabyte (PB) scale is required ranging, depending on the type of modeling, from 18 PB to 72 PB [25] (or even higher to handle the electrical synapses).

The goal of this study is to address the remaining two of the abovementioned key issues, namely, (1) morphology modeling of the human whole brain at the nanoscale, including somatic, dendritic, and axonal morphology, enabling the calculation of the synaptome (the set of all synapses) and connectome (the set of all neuronal circuits) and (2) designing a nanoscale human brain atlas, enabling the exploration of the human whole brain model.

## 2. Process of Nanoscale Atlas Creation

Navigating a macroscale brain atlas may not be a complicated task even when the atlas contains a few thousand components, like the atlas in [17]. However, navigating and exploring a nanoscale atlas will be enormously complicated, taking into account that, on average, the number of neurons in the human brain is about one hundred billion and the number of connections is one thousand trillion [26]. To deal with this data complexity, I propose to use a gross anatomy brain atlas at the macroscale, which is parcellated and labeled to facilitate the navigation of a nanoscale atlas. This approach will lessen the complexity and increase the confidence of the atlas user by employing a familiar neuroanatomy.

Moreover, any analysis can be simplified by restricting the explored volume of interest to some known structures, regions, nuclei, tracts, or functional systems.

A proposed process for the nanoscale human brain atlas creation is illustrated in Figure 1.

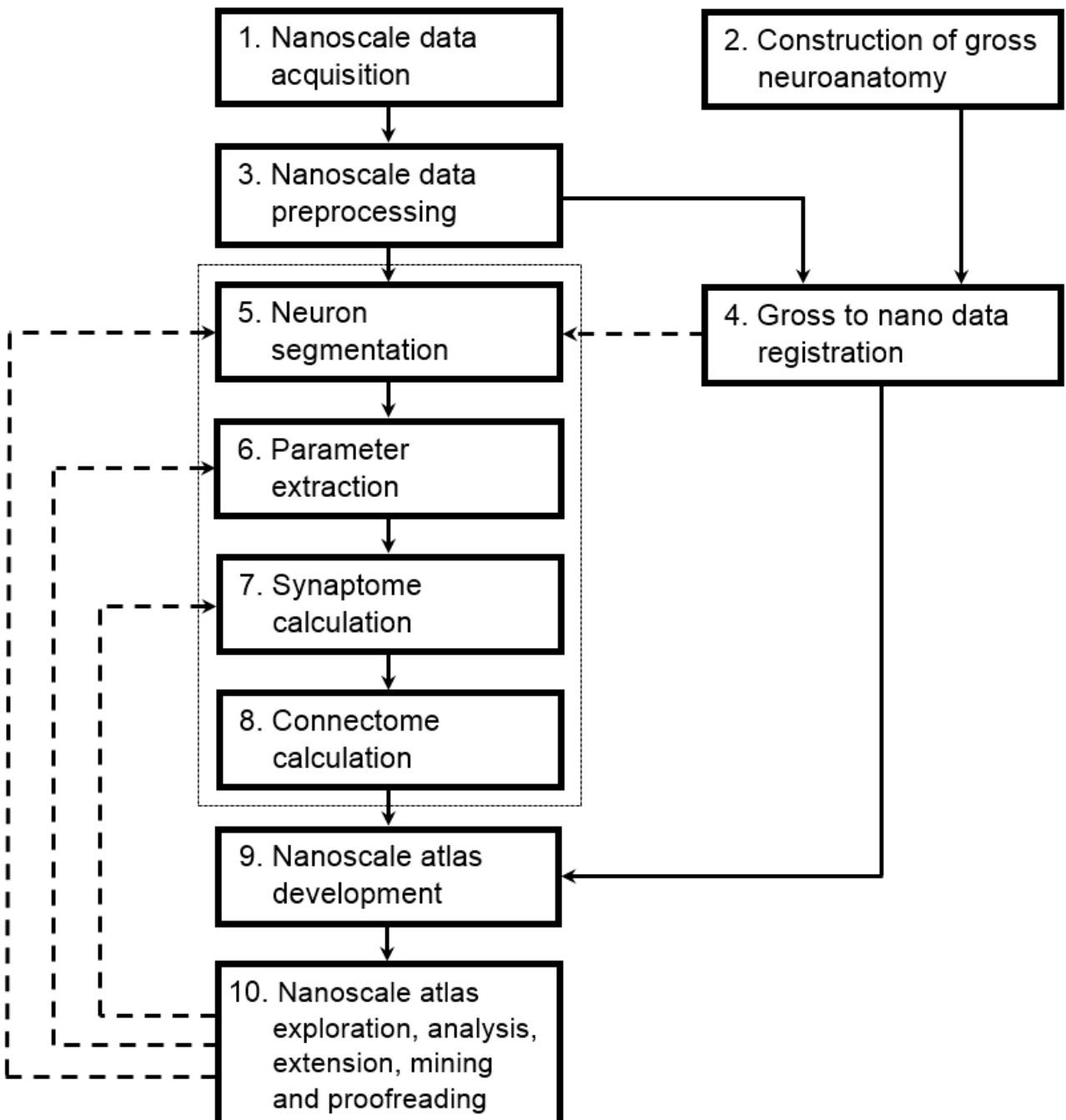

**Figure 1.** Overview of the process for the nanoscale atlas creation.

The nanoscale data need to be preprocessed and spatially registered with the gross neuroanatomy data. Preprocessing involves operations such as artifact removal, inhomogeneity correction, and image enhancement that are highly dependent on the process of data acquisition and the employed modality, and in the case of synchrotron tomography, these operations are addressed in [22].

The purpose of registration is to spatially align the macro and nano volumetric data. There are two broad categories of image reconstruction methods, feature-based

and intensity-based methods. Feature-based methods rely on some features, such as point or distributed landmarks, and, consequently, are not suitable for our purpose. Intensity-based methods rely on voxel intensities and to find a transformation that aligns them requires some similarity measure, such as cross-correlation, mutual information, or a sum of absolute or squared differences. Whole image content-based methods with numerous similarity measures are reviewed in [27–29]; moreover, *ITK* (Image Segmentation and Registration Toolkit) is a cross-platform, open-source development framework suitable for the implementation of registration applications [30]. In our case, a maximization mutual information-based method is suitable as it is useful for images of different modalities. It requires the nano volumetric data to be super-sampled so as to keep the cell bodies and the macro volumetric data to be sub-sampled to enable voxel-to-voxel comparison.

Next, the neurons have to be segmented from the nano volumetric data. Segmentation is modality dependent, and numerous methods have been developed for this purpose, such as region growing, thresholding with global or local thresholds and with or without thinning (skeletonization), edge detection with thinning, live wire, active contours (snakes), watershed, energy minimization, and deep learning. From the segmented neurons, suitable parameters shall be extracted, as defined in the nanoscale neuronal data format *nN* discussed below. Finally, the synaptome and the connectome shall be calculated.

The nanoscale data processing, encompassing step 5–8 (Figure 1), is computationally intensive. By decomposing each of these steps, this processing can be executed in a parallel-pipeline mode. Parallelization can be distinguished at two levels (1) across individual neurons with an extremely high processing granularity and (2) across sets of neurons in individual cerebral and cerebellar cortical regions and gray matter nuclei and sub-nuclei; this parallelization characterized by a medium granularity is feasible as the parcellated gross neuroanatomy registered with the nano neuroanatomy is available in step 4, Figure 1. The advantage of the latter is that it is additionally able to label the processed neurons with their identifiers (IDs) linked to gross neuroanatomy. These two types of parallelization exist in neuron segmentation, the calculation of neuronal parameters, and synaptome calculation. In the connectome calculation, a parallelization of a high granularity across the individual synapses of a processed neuron can be distinguished.

With a gross neuroanatomy model constructed and the nanoscale data processed, the nanoscale brain atlas can be developed. A brain atlas development requires determining design principles, content, architecture, navigation, functionality, and user interface. The main design principles for a macroscale brain atlas include top–down design, scalable cerebral model, and user interface, with an emphasis on esthetics. They result in an atlas that is 3D, interactive, fully parcellated, completely labeled, referenced, accurate, realistic, spatially consistent, stereotactic, multi-scale, extendable, composable, dissectible, explorable, user friendly, and modular. Two more general key principles are needed for the nanoscale brain atlas, namely, the handling of neurons, synapses, and neuronal circuits as well as the integration of macro with nano neuroanatomy.

An architecture of a multi-purpose and user-expandable brain atlas with multi-scale support has already been designed and its implementation is discussed in [13]. User expandability facilitates the nanoscale atlas proofreading and its potential extension directly by the user.

In terms of content, besides parcellated and labeled gross neuroanatomy in 3D, the nanoscale atlas shall contain the segmented neurons, synaptome, and connectome.

Navigation in a macroscale brain atlas, e.g., [31], enables fast group and/or structure selection to compose and decompose any 3D scenes for exploration. This navigation is facilitated by an anatomic index with double mapping between the index and the atlas content displayed in 3D. This principle is not suitable for neurons, synapses, and connectome due to their enormous numbers; thus, the navigation has to be extended, enabling a gross neuroanatomy-guided exploration of micro/nanoscale neuroanatomy. Neuroanatomy in macroscale atlases is typically labeled with names, whereas nano neuroanatomy shall be labeled with identifiers (IDs) of the neurons (besides their types and subtypes), synapses,



and neuronal circuits. Neurons shall additionally be labeled with the names of nuclei or cortical regions they belong to. Similarly, the neuronal circuits shall be labeled with the names of the white matter tracts they belong to.

The functionality of a macroscale brain atlas has been addressed in [31]. Having a 3D scene composed by the user, available main functions include manipulate (pan, rotate, zoom, set view), continuously navigate, dissect brain, read label, obtain stereotactic coordinates, measure, animate, and save the composed image. For the nanoscale neuroanatomy, labeling has to be extended as follows. First, a user shall be able to set a sampling volume of interest (sVOI). This sVOI may be fixed (e.g., assigned to a certain nucleus, such as the nucleus ambiguus in the brainstem) or be movable and scalable of a predefined shape, such as a cube or ball. Then, when sampling the nanoscale brain content with this sVOI, at its current location, the name of the corresponding nucleus or cerebral or cerebellar cortical region shall be given along with a list of neuron IDs within this sVOI and their afferent and efferent connections as well as the neuronal circuits intersecting this sVOI. When the sVOI is placed in white matter, then, the IDs of the neuronal circuits passing through it shall be provided along with the name of the corresponding white matter tract (or tracts), if available.

The exploration of nanoscale neuroanatomy will require a dedicated set of functions for morphometric and network-related analyses. These functions shall operate (1) globally for the whole brain; (2) locally for a selected nucleus or region of the cerebral or cerebellar cortex; and (3) in a user-defined sVOI. Morphometric functions shall provide the distributions of somata shapes and sizes, neuronal and synaptic densities, the thickness of cortical layers, and the distribution of cortical minicolumn, among others. Network-related functions shall provide various statistics, including the distributions of dendritic and axonal trees and the means for their comparison; the number of neuronal circuits and their distributions with respect to length, underlying structures, and tracts; local and global networks (subsystems); clusters (hubs), worldness (characterizing neighbors), betweenness centrality (measuring graph centrality based on shortest paths), and motifs (the recurring patterns of interconnections); loops and feedbacks; and the diverse patterns of interconnectivity.

Several user interfaces have been developed for neuroscience applications, such as the user interface of the *Allen Brain Atlas* that is under a continued process of standardization [1], a web-based user interface of the *Scalable Brain Atlas*, providing an instant access to public brain atlases and related content [32], a graphical user interface of *3D Slicer* supporting rich functionality [33], and the *Brainnetome Atlas* viewer with multiple views [34]. Here, we consider the user interface of *The Human Brain, Head and Neck in 2953 Pieces* atlas (shortly the *2953* atlas) [17,31], which is the result of our two-decade long efforts in designing and developing the brain atlas user interface, Figure 2. This user interface provides functions for display, navigation, exploration, and quantification of a 3D cerebral model. It supports two-way mapping between the model and the anatomical index as well as provides a continuous navigation of a 3D scene, which can be quickly composed and displayed in the main view. For handling the nanoscale neuroanatomy, the following extension of this user interface can be envisaged. The application shall work in two modes, macro (with meso) and nano (with micro), each with its own view. A zoom-in operation or an interactive switch shall swap the content in the main view from macro to nano. In the nanoscale neuroanatomy mode, the main view shall contain nano neuroanatomy, whereas macro neuroanatomy shall be moved to a supportive sub-view to guide navigation. The sVOI can initially be set against the macro neuroanatomy in this sub-view and subsequently fine-tuned within nano neuroanatomy in the main view. Having the sVOI set, it can be explored by obtaining labels (as discussed above) and performing morphometry functions for the sVOI content and network-related functions to encompass all connections from and to this sVOI.

Having the nanoscale atlas constructed can serve itself as a tool for subsequent interactive proofreading and neuron segmentation enhancement, parameter extraction, and synapse computing (Step 10, Figure 1).

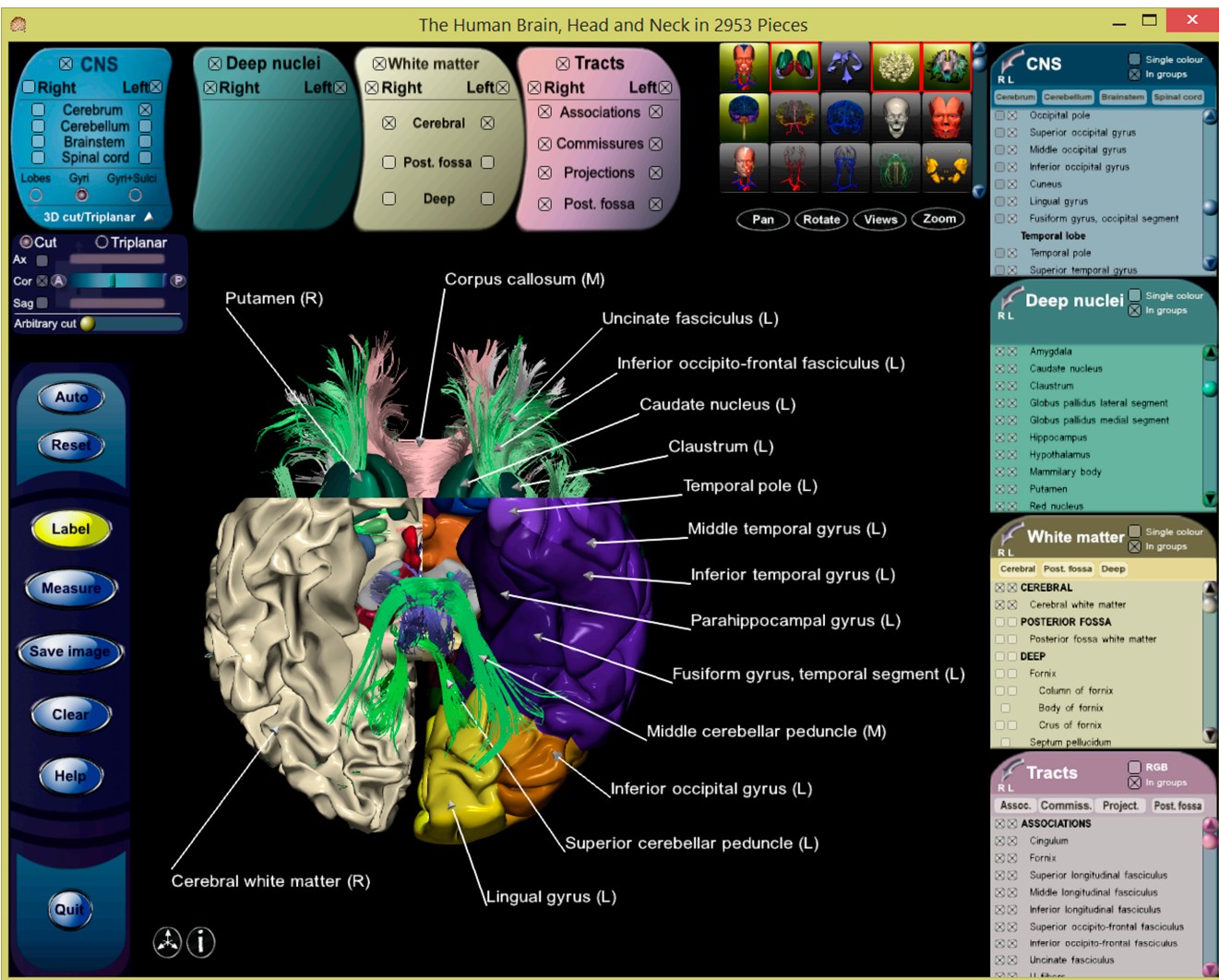

**Figure 2.** The user interface of the *2953* atlas. Four modules are selected and shown (viewed from the bottom of the brain) in the main view (center), the left cerebral hemisphere, the right cerebral white matter, the subcortical gray matter nuclei, and white matter tracts. The structures are parcellated by color and labeled with names. The left cerebral hemisphere with the cortex parcellated into gyri and the right cerebral white matter are dissected anteriorly to expose some white matter tracts and deep nuclei. The matrix with the content modules is on the top-right, whereas below it, there is the manipulation panel (whose functions are additionally mapped into the mouse buttons). The anatomical indices of the selected modules, each module with a vertically scrollable list of selectable structures, are shown in the panels on the right. The horizontally scrollable control panels of the selected modules are synchronized with their corresponding indices, each module providing the left/right side and group selections, and are on the top-left. The function panel is on the left.

## 3. Neuroanatomy Morphology Modeling

We consider neuroanatomy morphology modeling at the macroscale (cum the mesoscale) and the nanoscale (including the microscale).

### 3.1. Neuroanatomy Morphology Modeling at the Macroscale

There are two main representations for modeling cerebral structures in 3D, volumetric and geometric. A volumetric representation, for instance, has been exploited in the *VOXEL-MAN* brain atlas [16], providing semi-transparent volume rendering. A geometric representation has been employed in a family of five 3D atlases called the human brain in pieces, the latest being the *2953* atlas [17], as this representation enables creating models

at the sub-voxel level, enables interactive 3D scene composing and decomposing, and provides real-time interaction for complicated 3D scenes even on low-end computers and mobile devices [35].

Within geometric modeling, there are two key types of models, free-shape and tubular. Free-shape models, suitable to model, e.g., the cerebral cortex and the subcortical nuclei, can be reconstructed from neuroimages via iso-contouring by applying the Marching Cubes algorithm [36]. Tubular models enable the modeling of the arterial and venous systems as well as the cranial nerves. These structures, modeled either with circular or elliptical cross-sections, have variable radii or semi-axes, multiple branches, and their segmentation is error-prone, especially for small diameters. To cope with these issues, the modeling process involves some additional processing including the segmentation of tubular segments, the extraction of centerline and radius (or semi-axes), centerline processing by smoothing, radius processing by outlier removal and regression, the modeling of tubular segments, and bifurcations with circular or elliptical cross-sections by subdivision, tubular segment labeling, and color-coding [37].

### 3.2. Neuroanatomy Morphology Modeling at the Nanoscale

The human brain comprises neurons (nerve cells) for receiving, processing, conducting, and transmitting cellular signals and glial cells (neuroglia) that are supportive cells and supply oxygen and nutrients to the neurons; here, we only consider neurons. Neurons vary considerably in morphology, connectivity, electrophysiology, and molecular and genetic properties; here, we take into consideration their morphology. Numerous types of neurons can be distinguished based on morphology, such as multipolar neurons (e.g., pyramidal cells), bipolar neurons, basket cells, chandelier cells, and Purkinje cells, among many others. The total number of neuronal types is still unknown, and it could be as high as 1000 cell types [38]. Considering electrophysiology, transcriptome and projectome enable further neuronal classification and extension into neuronal subclasses, and a whole-brain cell atlas is under development [39].

Morphologically, a neuron contains soma (body, perikaryon) and neurites, which are projections from the soma. The neurites include multiple dendrites, which receive signals from other neurons and conduct information toward the cell body, and a single axon, which conducts a signal away from the cell body to other neurons [40]. Information is transferred from one neuron to another neuron at synapses, each synapse involving the presynaptic and postsynaptic neurons. The dendrites form a set of dendritic trunks (stems), each trunk with branches containing spikes with postsynaptic specializations with receptors termed here postsynaptic terminals. An axon emerges from the soma at the axon hillock, forming the axon proper (termed here axonal trunk), and at its distal end splits into branches, each ending with a terminal bouton (a presynaptic terminal) making contact with the next neuron.

Theoretically, the modeling methods employed for neuroanatomy at the macroscale can be applied to neuroanatomy at the micro- and nanoscales. Then, the free-shape modeling methods can be applied to model the soma, whereas the tubular methods can be employed to model the axons and dendrites. However, this solution is prohibitively expensive for the entire nanoscale brain model and shall rather be applied locally to facilitate exploring neuronal details.

As we focus on the neuronal morphology modeling at the nanoscale, a dedicated neuronal format is required. The neuronal format proposed here is an extended and enhanced version of a neuronal format introduced earlier [25]. The requirements for this format are (1) to comprise the complete dendritic and axonal trees without any simplification or reductionist encoding; (2) to embed gross neuroanatomy enabling nano neuroanatomy navigation; and (3) to include the dendritic and axonal terminals that determine the location and size, the latter if possible, of synapses.

The dendritic and axonal neurites can be considered the following trees

Root

- Nodes (bifurcations (generally, multifurfactions))
- Edges (branches)
- Leaves (terminal points)

Hence, the proposed nano neuronal format *nN* is the following:

**NEURON**

Neuron ID (n-ID)
Type of neuron
Subtype of neuron
Gross anatomy ID (nga-ID) neuron belongs to

**SOMA**

Center coordinates
Shape (predefined or free shape)

**DENDRITES**

Number of dendritic trunks
**For each** trunk

Trunk ID (dt-ID)
Proximal (at soma) coordinates
Proximal diameter
Dendritic tree root coordinates
Number of terminals in the dendritic tree
**For each** dendritic tree bi(multi)furcation

Dendritic tree bi(multi)furcation ID (dtb-ID)
Dendritic tree bi(multi)furcation coordinates
Dendritic tree bi(multi)furcation diameter

**For each** dendritic tree terminal

Dendritic tree terminal ID (dtt-ID)
Dendritic tree terminal coordinates
Dendritic tree terminal diameter

**AXON**

Axon ID (a-ID)
Hillock proximal (at soma) coordinates
Hillock proximal diameter
Axonal trunk proximal coordinates
Axonal trunk proximal diameter
Axonal tree root coordinates
Axonal tree root diameter
Number of bi(multi)furcations in the axonal tree
Number of terminals in the axonal tree
**For each** axonal tree bi(multi)furcation

Axonal tree bi(multi)furcation ID (ab-ID)
Axonal tree bi(multi)furcation coordinates
Axonal tree bi(multi)furcation diameter

**For each** axonal tree terminal

Axonal tree terminal ID (at-ID)
Axonal tree terminal coordinates
Axonal tree terminal diameter

It shall be noted that with sufficient spatial resolution, all the components of this format can be extracted from the segmented neurons excluding the dendritic tree terminal coordinates and diameters. The coordinates of the dendritic three terminal shall be calculated by processing the axonal tree terminals, such that for each axonal tree terminal, the corresponding dendritic tree terminal shall be localized in its immediate adjacency. However, the diameters of certain dendritic tree terminals may not exist depending on the shape of postsynaptic specialization regions.

As an example illustrating the *nN* format, consider a simple neuron schematically shown in Figure 3 with three dendritic trunks having, respectively, two, three, and six branches, and the axonal tree with five branches.

Let X denote (x, y, z) coordinates and D diameter. Then, the corresponding *nN* file is as follows (square brackets contain comments):

**[NEURON]**
    n1 [Neuron ID (n-ID)]
    pyramidal [Type of neuron]
    none [Subtype of neuron]
    precentral gyrus [Gross anatomy ID (nga-ID) neuron belongs to]
    **[SOMA]**
        n1(X) [Center coordinates]
        predefined, pyramid, scaling factor [Shape (predefined or free shape)]
    **[DENDRITES]**
        3 [Number of dendritic trunks]
        **For {1, 2, 3}** [**For each** trunk]
            {dt1, dt2, dt3} [Trunk ID (dt-ID)]
            {dt1, dt2, dt3}Proximal(X) [Proximal (at soma) coordinates]
            {dt1, dt2, dt3}Proximal(D) [Proximal diameter]
            {dt1, dt2, dt3}Dendritic tree root(X) [Dendritic tree root coordinates]
            {dt1, dt2, dt3}Dendritic tree root(D) [Dendritic tree root diameter]
            {0, 0, 2} [Number of bifurcations in the dendritic tree]
            {2, 3, 4} [Number of terminals in the dendritic tree]
            **For {1, 2, 3}** [**For each** dendritic tree bifurcation]
                {dtb31, dtb32} [Dendritic tree bifurcation ID (dtb-ID)]
                {dtb31(X), dtb32(X)} [Dendritic tree bifurcation coordinates]
                {dtb31(D), dtb32(D)} [Dendritic tree bifurcation diameter]
            **For {2, 3, 4}** [**For each** dendritic tree terminal]
                {dtt11, dtt12; dtt21, dtt22, dtt23; dtt31, dtt32, dtt33, dtt34} [Dendritic tree terminal ID (dtt-ID)]
                {dtt11(X), dtt12(X); dtt21(X), dtt22(X), dtt23(X); dtt31(X), dtt32(X), dtt33(X), dtt34(X)} [Dendritic tree terminal coordinates]
                {dtt11(D), dtt12(D); dtt21(D), dtt22(D), dtt23(D); dtt31(D), dtt32(D), dtt33(D), dtt34(D)} [Dendritic tree terminal diameter]
    **[AXON]**
        a1 [Axon ID (a-ID)]
        Hillock(X) [Hillock proximal (at soma) coordinates]
        Hillock(D) [Hillock proximal diameter]
        Axonal trunk proximal(X) [Axonal trunk proximal coordinates
        Axonal trunk proximal(D) [Axonal trunk proximal diameter
        Axonal tree root(X) [Axonal tree root coordinates]
        Axonal tree root(D) [Axonal tree root diameter]
        0 [Number of bifurcations in the axonal tree]
        5 [Number of terminals in the axonal tree]
        **For {0}** [**For each** axonal tree bifurcation]
            none [Axonal tree bifurcation ID (ab-ID)]
            none(X) [Axonal tree bifurcation coordinates]
            none(D) [Axonal tree bifurcation diameter]
        **For {1, 2, 3, 4, 5}** [**For each** axonal tree terminal}
            {at11, at12, at13, at14, at15} [Axonal tree terminal ID (at-ID)]
            {at11(X), at12(X), at13(X), at14(X), at15(X)} [Axonal tree terminal coordinates]
            {at11(D), at12(D), at13(D), at14(D), at15(D)} [Axonal tree terminal diameter]

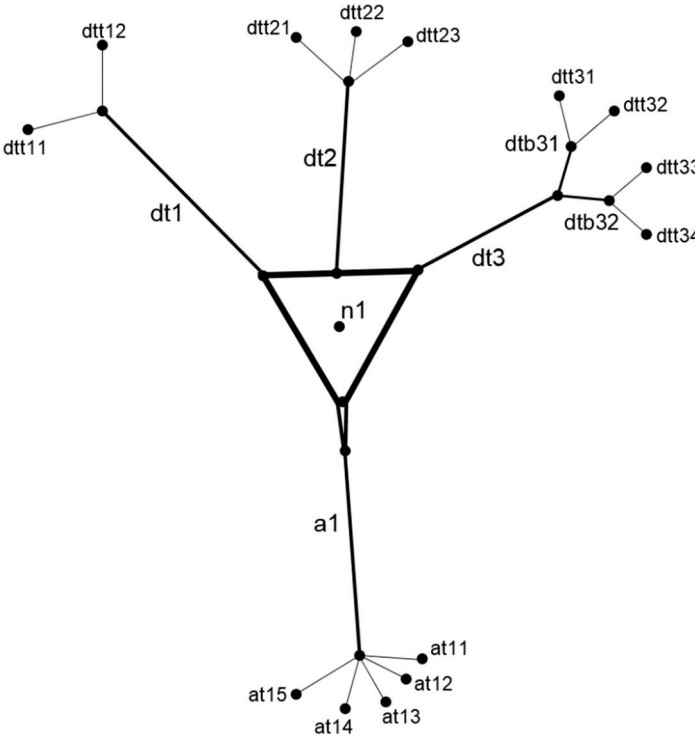

**Figure 3.** A simplified model of a neuron to illustrate the nano neuron data format. The neuron, axonal and dendritic trunks, bifurcations, and terminal points are labeled with their identifiers. The big dots mark the locations where the coordinates and diameters are measured.

## 4. Discussion

Tremendous progress in neuroimaging and high-performance computing makes the acquisition of volumetric micro/nano datasets, covering the entire human brain, and the development of a whole brain atlas at the nanoscale feasible in the non-distant future. What is practically possible today is to relax the requirement of brain entirety and acquire a small nanoscale brain sample. For instance, in [41], a human surgical tissue sample of a 1 mm$^3$ volume was collected and then stained, embedded in resin, cut into ~30 nm thick sections, and scanned using electron microscopy. The sample contained about 19 thousand reconstructed neurons and 133.7 million synaptic connections (note that despite a nanoscale sectioning, the data were not sufficiently sampled according to the Nyquist theorem missing about one-third of connections). Toward the development of a human whole brain atlas at the nanoscale, we here address two challenges, i.e., the morphology modeling of the whole brain at the nanoscale and designing of a nanoscale brain atlas.

Several data formats have already been developed to store, process, model, analyze, quantify, compare, classify, and share neuronal morphology, including *Eutectic* [42], *Neurolucida* [43], and *SWC* [44]. *SWC* is a widely used format of neuron morphologies due to its simplicity and the short representation of the neuronal arbor branching skeleton, which makes it sufficient for most research applications [45]. Its formal specification is available at https://swc-specification.readthedocs.io (accessed on 22 November 2023). Several *SWC* variants have also emerged including *SWC+* to allow lines, contours, images, and surfaces to be embedded. The *SWC* format has been adopted by *NeuroMorpho.Org*, which is to date the largest freely accessible database of neuronal and glial digital reconstructions [46]. *NeuroMorpho.Org* contains more than 140,000 reconstructions and provides automatic calculation of 21 morphometric parameters for each cell, including soma surface, number of branches, length, volume, angles, topological asymmetry, fractal dimension, and taper rate [47].

For simulation purposes, models and tools have also been developed to generate virtual neurons with the required distributions of neuronal parameters, including total dendritic length, total surface area, topological asymmetry, total number of bifurcations,

maximum branching order (number of bifurcations to soma + 1), average path (along the dendrite) from soma to tips, average distance from soma to tips, maximum distance from soma to tips, and significant (contains 95% of segments) height, width and depth [48].

To facilitate and expedite the comparison and classification of neurons, neuronal tree encoding is applied, including *Persistence Diagram Vectors* and *Topological Morphology Descriptor*. *Persistence Diagram Vectors* compactly describe the branching patterns of neuronal and glial trees considering the distributions of the number of branches as a function of distance from the soma [49]. *Topological Morphology Descriptor* encodes the branching pattern of the morphology by discarding local fluctuations with little information content, such as the position of the nodes between branch points, and thus reduces the computational complexity of a tree while encoding the overall shape of the tree [50].

Several software packages have been developed for neuron visualization, such as *Neurolucida* [43] and a web-based *HBP Neuron Morphology Wiewer* [51], as well as for modeling and simulation comprising *NEURON* [52], *TREES Toolbox* [53], and *GENESIS* [54]. Moreover, an international community effort the *BigNeuron* project (https://alleninstitute. org/news/bigneuron-project-launched-to-advance-3d-reconstructions-of-neurons/ accessed on 22 November 2023) defines and advances the state-of-the-art of single-neuron reconstruction from optical microscopy images, develops a toolkit of standardized reconstruction protocols, analyzes neuron morphologies, and establishes a data resource for neuroscience [55].

The existing and popular neuronal morphology data file formats differ from the nano neuronal *nN* format introduced here in terms of purpose, content, and structure. The existing file formats, such as *SWC*, are neuroinformatics resources for tracing, visualization, analysis, simulation, and comparison of neurons and neuronal trees, and they are not fully sufficient to enable the construction of a nanoscale human brain model. The *nN* format captures the complete data for morphology modeling of the entire human brain, enabling the calculation of its synaptome and connectome. It should be emphasized that the knowledge of the human connectome is vital for understanding how neuronal circuits encode information and how the brain works in health and disease [4]. In terms of content, the existing formats comprise multiple various parameters, such as the number of branches, total dendritic length, total surface area, angles, topological asymmetry, fractal dimension, taper rate, average and maximum distances from soma to tips, and significant height, width and depth. However, they do not contain the presynaptic and postsynaptic terminals and, consequently, the synaptome. On the other hand, the *nN* format has just two data types, points and diameters, making its structure simple, and the knowledge of these points is sufficient to compute the connectome. The *nN* format does not store neuronal topology (saving in this way a substantial space); although crucial in neurite tree comparison and available in the existing formats, topology is unneeded for the connectome calculation (however, if in some applications, like neurite tree comparison, a "topologylessness" of the *nN* format would be a problem, *nN* can be augmented with an *SWC*-type of format). Finally, the *nN* format links macro with nano neuroanatomy. In terms of application, the *nN* format has been employed to estimate big data to store nanoscale neuroanatomy [25] and high-performance computing resources to calculate it [56].

The *nN* format has a certain limitation resulting from data economization. It assumes that a neurite branch is represented by two points only meaning it is a linear section. This simplification does not capture the full geometry of the dendritic and axonal branches but is sufficient to compute the connectome. Obviously, it is possible to introduce intermediate points at the expense of increasing storage as discussed in [25], which may be especially crucial for long axons. Introducing one additional point per branch makes the branches parabolic but still planar. Two additional points enable cubic branches. Moreover, the synapse modeling is limited to the presynaptic and postsynaptic terminal points with diameters, and though simplified, it is sufficient to determine a synapse. However, a synapse contains subsynaptic structures, including bouton, active zone, postsynaptic density, and dendritic spine, which are highly correlated in their dimensions and

correlate with synapse strength as well as they also change during episodes of synaptic plasticity [57]. Future extensions of the *nN* format shall accommodate for that.

Human brain atlases have been tremendously developed in terms of content, functionality, and applications [8]. There are numerous methods and tools for the construction of brain atlases, and they have extensively been discussed and reviewed earlier [13,58]. Several atlases have been built for the human whole brain from neuroimaging, such as *Digital Anatomist* [14], atlas for education, segmentation, and surgical planning [15], *VOXEL-MAN* [16], and *The Human Brain, Head and Neck in 2953 Pieces* (the *2953* atlas) [17]. Volumetric atlases, such as VOXEL-MAN, do not allow for the selection of individual structures or their groups; thus, they are not suitable for guiding navigation in a nanoscale atlas. The *2953* atlas has been designed to rapidly compose any 3D scene that is fully parcellated and labeled, and this kind of atlas is suitable for navigation. Although it contains about 3000 3D pieces, for a macro/meso-guided navigation, it shall be further extended with other atlases, providing the finer parcellations of deep nuclei (see below), cerebral and cerebellar cortical regions, white matter tracts, and functional systems.

To create a nanoscale brain atlas, I have proposed here three novel atlas design principles supporting navigation, exploration, and calculations, respectively, namely, (1) the navigation of micro/nanoscale neuroanatomy guided by a macro/meso parcellated and labeled neuroanatomy atlas forming a reference map; (2) a movable and zoomable sampling volume of interest (sVOI) for navigation and the subsequent exploration of neurons, synapses, and neuronal circuits in this sVOI; and (3) a nanoscale data processing in a parallel-pipeline mode exploiting parallelism resulting from the decomposition of gross neuroanatomy parcellated into structures and regions as well as nano neuroanatomy decomposed into neurons and synapses, enabling the distributed construction and continual enhancement of the nanoscale atlas.

The first principle may facilitate navigating the enormous networks of neuronal connections without the clear patterns of organization and visible landmarks by providing a macro/meso localizer. Its effectiveness depends on the level of parcellation of the macro/meso brain atlas. The finer the parcellation, the easier the guidance of a sVOI placement, see Figure 4.

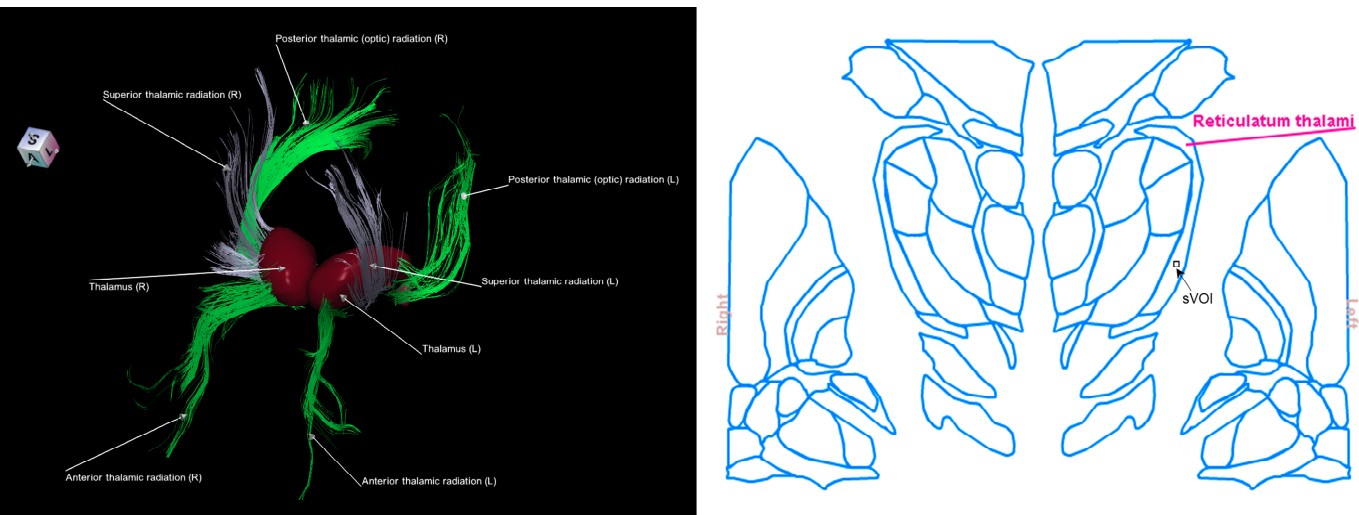

**Figure 4.** (**Left**) The entire thalamus along with its cortical projections labeled with names (from the *2953* atlas). (**Right**) A coronal section through the thalamus from *The Cerefy Clinical Brain Atlas* [59] (with permission from Thieme) with a finer subdivision of the thalamus based on the Hassler parcellation (who distinguished more than 100 thalamic nuclei [60]). For illustration, sVOI of 0.4 mm size is set in the reticulatum thalami for the nanoscale atlas to demonstrate all afferent and efferent connections in this region.

The sVOI works both as a sampler and labeler. A movable and zoomable sVOI makes the navigation in a nanoscale atlas Google map-like. Another advantage of employing the sVOI is that the restriction of the explored region in the nanoscale atlas to this sVOI content facilitates interactivity in contrast to the whole nanoscale atlas whose handling requires high-performance computing.

In contrast to the development of a macroscale brain morphology model that at the nanoscale requires big data and high-performance computing, and it shall be rather a multi-centered effort, particularly, since each of the neuron segmentation, neuronal parameter extraction, synaptome calculation, and connectome computing steps can be decomposed and performed concurrently. After completing morphology, we can envisage future directions consisting of adding electrophysiology, molecular components, glial cells, and microvasculature as well as creating a probabilistic nanoscale atlas by employing multiple instances of the *nN* file format.

Numerous applications of the nanoscale human brain atlas can be contemplated ranging from proofreading and continual extension at multiple sites to exploration, morphometric and network-related analyses, and knowledge discovery.

To my best knowledge, this is the first proposed neuronal morphology model at the nanoscale and the first attempt to design a human whole brain atlas at the nanoscale.

**Funding:** This publication is supported by the European Union's Horizon 2020 research and innovation programme under grant agreement Sano No. 857533. This publication is supported by Sano project carried out within the International Research Agendas programme of the Foundation for Polish Science, co-financed by the European Union under the European Regional Development Fund.

**Data Availability Statement:** No dataset was used or generated. The presented approach is freely available.

**Acknowledgments:** This study has been inspired by the SYNAPSE (Synchrotron for Neuroscience—an Asia-Pacific Strategic Enterprise) efforts, aiming to map the entire human brain at sub-cellular resolution, and some ideas presented here are extensions of my invited talk given at the SYNAPSE 2022 general meeting.

**Conflicts of Interest:** The author declares no conflict of interest.

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
