# Peer review of "Toward Morphologic Atlasing of the Human Whole Brain at the Nanoscale"

_2504-2289, doi:10.3390/bdcc7040179_

Round 1

Reviewer 1 Report

Comments and Suggestions for Authors

Comments on the Quality of English Language

To enhance overall readability and coherence, it is recommended to make concise and logically structured revisions throughout the text. Specifically, repeated statements of the same ideas should be avoided.

Reviewer 2 Report

Comments and Suggestions for Authors

In this study, the author discussed the potential construction of a nanoscale atlas for the entire human brain, addressing challenges in nanoscale brain morphology modeling and atlas design. The introduction of a new nanoscale neuronal format was presented as a solution for modeling at this level, allowing for synaptome and connectome calculations. The design principles of the nanoscale brain atlas were outlined, emphasizing navigation, exploration, and calculations. It claimed to be the first proposal for a nanoscale model of neuronal morphology and a nanoscale whole human brain atlas. While the study outlined ambitious goals and innovative approaches, here are few concerns about the proposed methods.

1.     The author discussed the challenges of navigating a nanoscale brain atlas due to the complexity of dealing with a vast number of neurons and connections. The use of synchrotron X-ray tomography was suggested as a promising imaging modality, but concerns about practicality and accessibility may arise. Also, the enormity of nanoscale data and the proposed tasks of preprocessing, segmentation, and spatial registration with gross neuroanatomy could pose significant computational challenges. Does the author has any thought about the processing techniques?

2.     The proposed neuroanatomy modeling discusseD macro and nanoscale approaches. While it recognizes existing formats like SWC, the new nN format lacks a detailed comparison. Additionally, the economization in representing neurite branches simplifies the model but lacks exploration of its impact. Comparisons with existing models and atlases might provide a more balanced evaluation of the proposed advancements.

Round 2

Reviewer 1 Report

Comments and Suggestions for Authors

While the scope of this study does not involve direct data acquisition, the inclusion of an example with real data is crucial to elucidate the processed neurons and their corresponding identification codes intricately linked to gross neuroanatomy. This illustrative example serves as a pivotal aid, providing readers with a comprehensive insight into the intricacies of the proposed nanoscale atlas design and facilitating a nuanced understanding of the methodology employed for data mining and subsequent analysis.

Reviewer 2 Report

Comments and Suggestions for Authors

My questions have been addressed in the revised manuscript.